# Practices and trends in clinical trial registration in the Pan African Clinical Trials Registry (PACTR): a descriptive analysis of registration data

Duduzile Edith Ndwandwe ![ORCID], Sinazo Runeyi, Elizabeth Pienaar, Lindi Mathebula ![ORCID], Ameer Hohlfeld, Charles Shey Wiysonge ![ORCID]

Cochrane South Africa, South African Medical Research Council, Cape Town, Western Cape, South Africa

**Correspondence to**
Dr Duduzile Edith Ndwandwe; duduzile.ndwandwe@mrc.ac.za

## ABSTRACT

**Background**  The Pan African Clinical Trials Registry (PACTR) is a WHO International Clinical Trials Registry Platform primary register, which caters for clinical trials conducted in Africa. PACTR is the first and, at present, the only member of the Network of WHO Primary Registers in Africa. The aim is to describe and report on the trends of trial records registered in PACTR.

**Methods**  PACTR was established in 2007 as the AIDS, Tuberculosis, and Malaria Clinical Trials Registry. The scope of the registry was then expanded in 2009 to include all diseases. This is a cross-sectional study of trials registered in PACTR from inception to 18 August 2021. A descriptive analysis of the use and trends of the following data fields: study intervention, disease condition, sex of the participants, sample size, ethics, funding and availability of results was conducted using Microsoft Excel.

**Results**  The number of trials registered has increased year on year, reaching 606 trials registered in 2020. The total number of trials registered at the time of the analysis was 2998. More than half of the trials in the registry (1655 of 2998, ie, 55%) were prospectively registered. Ethical approval was received by 90% (2691 of 2998) of the registered trials. Factorial assignment as an intervention model was in 20% (589 of 2998) of the trials registered. There were 36% (1083 of 2998) completed trials, of which 3% (94 of 1083) had results available in the registry. The most dominant funding source indicated was self-funding in 23% (693 of 2998) of the registered trials, and 55% (1639 of 2998) had no funding.

**Conclusion**  Registration on PACTR continues to grow; however, our analysis shows that researchers' capacity-building is needed to understand the importance of the registry and how this information informs healthcare decisions. Promoting prospective trial registration remains critical to avoid selective reporting bias to inform research gaps.

## INTRODUCTION

The Pan African Clinical Trials Registry (PACTR) (www.pactr.org) was established from the AIDS, Tuberculosis, and Malaria Registry based at the South African Cochrane Centre.[1–3] The registry was established with Cochrane's Infectious Diseases Group, based

### Strengths and limitations of this study

► We provided a comprehensive descriptive assessment of the trials registered in the Pan African Clinical Trials Registry (PACTR).
► We conducted a descriptive analysis to assess the trends of the fields collected in the registry records to improve and prioritise activities for PACTR administration staff.
► We selected mandatory data fields to analyse to precisely assess the general trends in the trial records without analysing the free-text data captured.
► There were some unavoidable missing data and variations for certain data fields, which might bias the results.

at the Liverpool School of Tropical Medicine and the WHO. PACTR is the only African member of the WHO Network of Primary Registers, which transfers trial information to the WHO International Clinical Trials Registry Platform (WHO-ICTRP) (https://www.who.int/clinical-trials-registry-platform) every month.[4 5] WHO-ICTRP serves as a platform aligned with the International Committee of Medical Journal Editors for prospective trial registration. PACTR contributes to regional transparency and harmonisation of clinical trial research[6 7] and is freely available. A database contains essential administrative and scientific information about planned, ongoing and completed trials in a clinical trials registry.[6–8] Thus, registration of all interventional trials is considered scientific, ethical and responsible.[9–11] Accessing clinical trials information allows informing decision-making on healthcare decisions based on all available evidence.[9] Such decisions cannot be easily made if publication bias and selective reporting exist.[9]

Furthermore, the Declaration of Helsinki indicates that 'Every clinical trial must be

registered in a publicly accessible database before recruitment of the first subject'.[9] In the case of clinical trials, before the first participant is recruited, the information on the trial must be captured in a publicly accessible database unless the sponsor or researcher has permission to delay this to a later stage. Trial registration is one of the efforts being made to ensure transparency in clinical research, accessible patient data for subsequent analysis and publication of results irrespective of the trial outcome. This further allows for decisions related to the safety and efficacy of drugs, vaccines and medical devices in humans, supported by the best available scientific evidence.

This, therefore, implies that clinical trial registration should advocate for prospective trial registration and that all registered trials publish their findings.[12] Trial registration further supports evidence-based medical practice, which heavily relies on available data in the public domain so that informed healthcare decisions can be made.[13] Bringing in data from clinical trials within reach of clinicians, regulators and external stakeholders enhances the clinical trial data.[13] Prospective trial registration and subsequent results reporting are global efforts to ensure complete research transparency. Clinical trials may be registered without ethics approval, provided that recruitment of study participants has not commenced. Even journal editors, ethics committees/institutional review boards, regulatory authorities and funding agencies all support the call for research transparency requiring trials to be prospectively registered.[14]

There has been a push from governments and international organisations, especially since 2005, to make clinical trial information more widely available and standardise registries and registering processes. The WHO has published international Standards for Clinical Trial Registries to achieve consensus on both the minimal and the optimal operating standards for trial registration.[14] To adhere to WHO practices that ensure that collected data are not duplicated and provide meaningful information, registry staff scrutinise each application and perform regular quality checks to ensure quality data are contained in the registry.

A further benefit to registering trials prospectively in a registry is that it allows for similar or identical trials to be known, making it possible for researchers and funding agencies to avoid unnecessary duplication.[7] Also, describing clinical trials in progress makes it easier to identify research gaps for new research to advance the knowledge gaps. Registries provide quality checks on the data submitted as part of the registration process, leading to improvements in the quality of clinical trials publicly available and pointing out potential problems early in the research design to improve clinical research conducted.

Although in the past, research on the clinical trial landscape provided key insights into the global burden of disease, and more generally, the global and regional clinical trial landscapes,[4 7 15 16] before PACTR, there was no regional support for longitudinal monitoring of planned and ongoing African clinical trials. PACTR is unique in recognising that African researchers face additional challenges in trial registration and seeks to provide feasible ways of overcoming these barriers.[2] PACTR has seen substantial growth in the number of trials registered from inception until recently. In this cross-sectional survey of the PACTR database, we report on the trends in the clinical trials registered.

## METHODS

This was a descriptive analysis of the trends in clinical trials registered in the PACTR (www.pactr.org).

### Data description and source

We used the WHO-ICTRP (https://www.who.int/clinical-trials-registry-platform), a registry platform collating information from registries across the globe to be a one-stop portal to access clinical trial records.[17] The study used the WHO definition of a clinical trial: 'any research study that prospectively assigns human participants or groups of humans to one or more health-related interventions to evaluate the effects on health outcomes'.[14] We used the advanced search function of ICTRP to identify these clinical trials registered in the PACTR on 18 August 2021.

### Data management and analysis

Data were downloaded from WHO-ICTRP by one researcher (SR) on 18 August 2021 and exported into an Excel spreadsheet. All records were quality checked by a second researcher (DEN). In each record, the following data items were used for analysis: registration status, disease condition, sex of the participants in the trials, sponsor, intervention type, funding source, the age range of participants, intervention model, phase of the trial and overall status. We conducted a descriptive analysis of the use and trends of the registered trials in PACTR to understand the pattern of trial registration over the years using Microsoft Excel.

### Patient and public involvement

Patients and/or the public were not involved in the design, or conduct, or reporting, or dissemination plans of this research.

## RESULTS

We report on the trends for trials registered in PACTR appearing in the ICTRP portal. PACTR is one of the WHO primary registers which sends data monthly to the ICTRP for one-stop access to trial records. We downloaded from the ICTRP on 18 August 2021. We used the search output to only select trials from the PACTR. A total of 2998 trial records were retrieved and used for analysis.

PACTR has grown substantially since its inception, with each year showing a steady increase in the number of trials registered. The year 2020 had the most registered trials (n=606). We anticipate that this increase will be seen even in 2021 (figure 1).

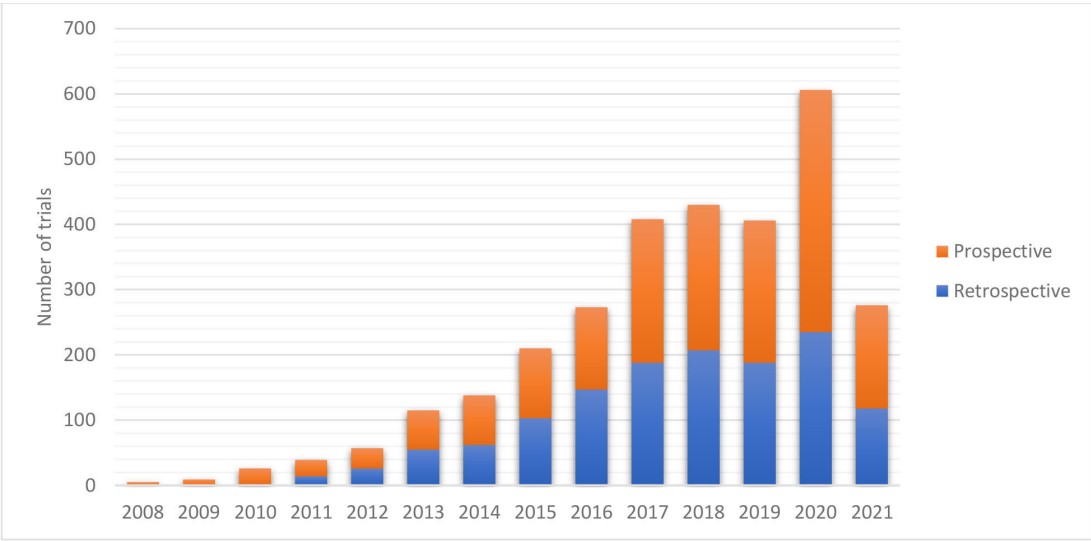

**Figure 1** Number of retrospective and prospective registrations on PACTR by year. We conducted a descriptive analysis of the trials registered in PACTR and showed the number of trials registered per year on the x axis. The orange bar represents trials flagged as prospective registration, and the blue bar represents trials flagged as retrospective upon registration. The y axis represents the number of trials. PACTR, Pan African Clinical Trials Registry.

We further extrapolated the trials registered in 2020 to assess whether the significant increase was because of the COVID-19 pandemic, which has seen a rise in research activity.

We found that 7% (42 of 606) were COVID-19-related trials in the year 2020 and among these trials, 46% were on treatment and 20% on vaccines (figure 2).

Table 1 shows our analysis of some of the registry data items. Generally, there has been an increase in the number of trials registered in PACTR, with n=2998 identified at the analysis time (figure 1). There are 1083 (36%) completed trials, with 94 (3%) having results available in the registry. Twenty-eight per cent (28%) of trials registered (836 of 2998) are listed as not recruiting, while 25% (755 of 2998) are recruiting participants. Fifty-five per cent (1655 of 2998) of the trials are registered prospectively, with the remaining 45% (1343 of 2998) registered retrospectively.

Our data show that most of the trials registered in PACTR have ethics approval (2691 of 2998, ie, 90%). The intervention model refers to the general design of the strategy for assigning therapies or interventions being investigated to participants in a clinical study. Types of intervention models include single group assignment, parallel assignment, cross-over assignment and factorial assignment. The most common intervention model in the registered trials was factorial assignment (589 of 2998, ie, 20%), which means that the trial would have two (or more) intervention comparisons carried out simultaneously. The trial phases show an almost equal distribution for all clinical trial phases. We assessed the sponsor of the registered trials and reported that the sponsor could be the funder. Our data show that 55% (1639 of 2998) of the trials have no funding, while 23% (693 of 2998) are self-funded. Many trials (2240 of 2998, ie, 75%) recruited

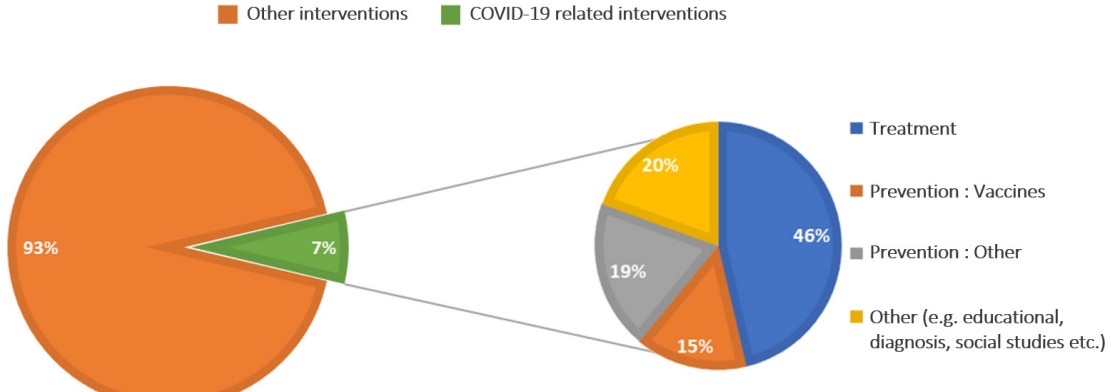

**Figure 2** An assessment of the trials registered in the year 2020. We describe the number of trials registered in 2020 presented as a pie chart indicated in orange and green colours. The green pie represents the COVID-19 trials which are further expanded to show the different interventions of these trials in different colour shades.

| Table 1 Characteristics of trials in PACTR | |
|---|---|
| **Description** | **N (%)** |
| Number of trials registered | 2998 |
| Number of studies completed | 1083 (36.1) |
| Number of completed with results | 94 (3.1) |
| Overall trial status | |
| Completed | 1083 (36.1) |
| Recruiting | 755 (25.2) |
| Not yet recruiting/pending | 836 (27.9) |
| Recruiting | 755 (25.2) |
| Stopped/terminated | – |
| Suspended | 7 (0.2) |
| Pending | 836 (27.5) |
| Other/unknown | 317 (10.6) |
| Prospective/retrospective | |
| Prospectively registered | 1655 (55.2) |
| Retrospectively registered | 1343 (44.8) |
| Intervention model | |
| Parallel assignment | 2124 (70.8) |
| Single group assignment | 59 (2.0) |
| Cross-over assignment | 201 (6.7) |
| Factorial assignment | 589 (19.6) |
| Sequential assignment | 13 (0.4) |
| None (open label) | 12 (0.4) |
| Phase | |
| Not reported | 2310 (77.1) |
| Phase I | 192 (6.5) |
| Phase II | 138 (4.6) |
| Phase III | 199 (6.6) |
| Phase IV | 155 (5.2) |
| Primary sponsor | |
| University | 196 (6.5) |
| Industry or non-governmental organisation | 61 (2.0) |
| Government | 107 (3.6) |
| Charities | 94 (3.1) |
| Hospital | 67 (2.2) |
| Self-funded | 693 (23.1) |
| Funding agency | 142 (4.7) |
| Other | 45 (1.5) |
| No funding | 1639 (54.7) |
| Ethics approval received | |
| Yes | 2691 (90) |
| No | 307 (10) |
| Sex | |
| Both male and female | 2240 (74.7) |
| Female | 628 (20.9) |
| | Continued |

| Table 1 Continued | |
|---|---|
| **Description** | **N (%)** |
| Male | 130 (4.3) |
| The target number of participants | |
| Minimum, maximum | 0–1 087 000 |
| Mean (SD) | 1–140.7 (26–156.8) |
| Median (IQR) | 80 (1–125) |

PACTR, Pan African Clinical Trials Registry.

both male and female participants. The median sample size was 1140.7 participants, ranging from 0 to 1 087 000.

Researchers have an option in the PACTR to indicate the type of intervention for the trial. We show that most common intervention type specified was drug treatment (622 of 2998, ie, 21%; figure 3).

The most common disease conditions investigated in the trials conducted in PACTR-registered trials were infections and infestations with 20% (586 of 2998), followed by the surgery category with 14% (426 of 2998) trials. The trials listed surgery as a disease condition included any intervention in a trial where medical and surgical care was provided. Such studies focus on diseases, injuries, and conditions affecting the abdomen, breasts, digestive system, endocrine system, and skin. Also, these trials evaluate biopsies, lab tests and imaging tests as part of delivering care (figure 4).

Among the completed trials (n=1083), most of the records are without results (91%; 989 of 1083), and less than 10% have results (figure 5).

The reporting section in PACTR was not mandatory until 2019. Our data show that from 2008 until 2017, results reporting was not captured. In 2018, PACTR was relaunched to include the 24-item data set required by ICTRP.[14] The reporting section is the 24th data item which collects information on the plans to share trial data and provides summary results. When PACTR was relaunched in 2018, this field was not mandatory and had options 'yes', 'no' and 'undecided'. We, therefore, assessed whether the trials in PACTR reflected the occurrence of adding the additional data fields.

The data show that 4% (114 of 2998) of researchers opted to choose undecided when capturing the trial information, while 7% left this section blank (222 of 2998). In 2019 when the 'undecided' option was removed, and the field became mandatory, there was a shift in the trend with 11% (342 of 2998) indicating 'yes' to complete the results reporting section. This trend continues with more trials in 2020 and 2021, opting for completing the reporting section (figure 6).

## DISCUSSION
PACTR has seen a growing number of trials registered over the years. We, therefore, report on trends in the registration of clinical trials in PACTR. Understanding the

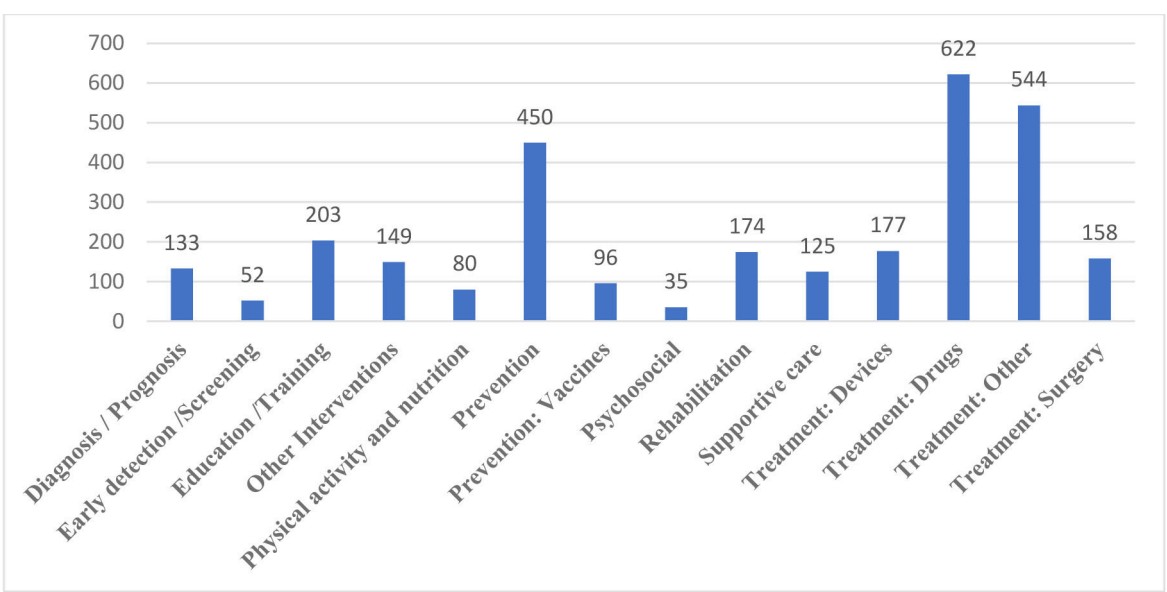

**Figure 3** Type of intervention used for the registered trials in PACTR. The trials registered indicate the intervention being investigated in their record. We describe the intervention of all the trials registered at the time of analysis. The results are represented as a bar graph indicating the number of trials with a specific intervention. The total number of trials for a particular intervention is presented at the top of each bar. PACTR, Pan African Clinical Trials Registry.

trends will allow for further improvements on the registry and identify issues that the registry team can improve on. Our data show substantial growth in the number of registered trials over the years. PACTR registered 606 trials in 2020, contributing to 20% of the analysed trials. COVID-19-related trials only contributed to 7% of the increasing trials registered in 2020. Among these trials, 46% (19 of 41) being investigated, the most dominant intervention was treatment for COVID-19. This trend is on an upward trajectory as even in the year 2021, there are 15 trials related to COVID-19.[18]

In our efforts to encourage prospective registration of trials, a slight shift towards prospective trial registration can be seen from 2017. Prospective trial registration is currently at 55%, while retrospective trial registration is 45% (figure 1). Trials can be registered retrospectively; however, the prospective registration of trials is encouraged to ensure transparency in research conduct, thus reducing publication and reporting bias.[9] Efforts to register trials prospectively need to be done across all primary registers. Al-Durra et al conducted a cross-sectional analysis of published trials registered in registries worldwide and found that prospective registration is deficient.[9] PACTR allows a trial registration if the researcher indicates when ethics approval has been applied for. We show that among the trials registered in PACTR, 90% have ethics approval which shows that the trials conducted have gone through the ethics approval process. PACTR staff also ensure that the ethics approval is verified to ensure that the data in the registry are correct.

The intervention model in PACTR-registered trials indicates that registries may need to adapt to the changing trial designs, as seen with the current COVID-19 trials

where adaptive trial designs were used.[19–21] Our analysis shows that the most common intervention model was factorial assignment (20%). The studies registered in PACTR show a worrying trend which shows that 55% of the trials have no funding while 23% of the trials are self-funded. Similarly, a recent cross-sectional bibliographical study showed that tuberculosis trials conducted in Africa had a dearth of financing for local African governments and non-governmental organisations.[22] There should be a shift for African governments and funders to create appropriate ways to ensure that total costs of clinical research are provided. Research institutions and universities with a real potential for success should have priority so that resources can be focused on driving research programmes for Africa.[23]

The other concerning trend from our analysis is that 28% of the trials are listed as not recruiting. This is indicative that researchers do not update the records, which could result in the data being misinterpreted. The 'not recruiting' status indicates that participants are still receiving an intervention or being examined, but new participants are not currently recruited or enrolled. However, it may also suggest that this status indicates that all participant visits are completed, the study is still open to ethics, data analysis is still ongoing or the manuscript is pending publication. This suggests a need to build capacity on the 24-item data set.[14]

Moreover, capacity building should focus on the vital role of registries as a source of data sharing, identifying research gaps and its essential contribution in the evidence ecosystem,[23] rather than another administrative activity to conduct their trials. Our data show that of the completed trials in the registry, only 3% have results available. This suggests that PACTR needs to partner with

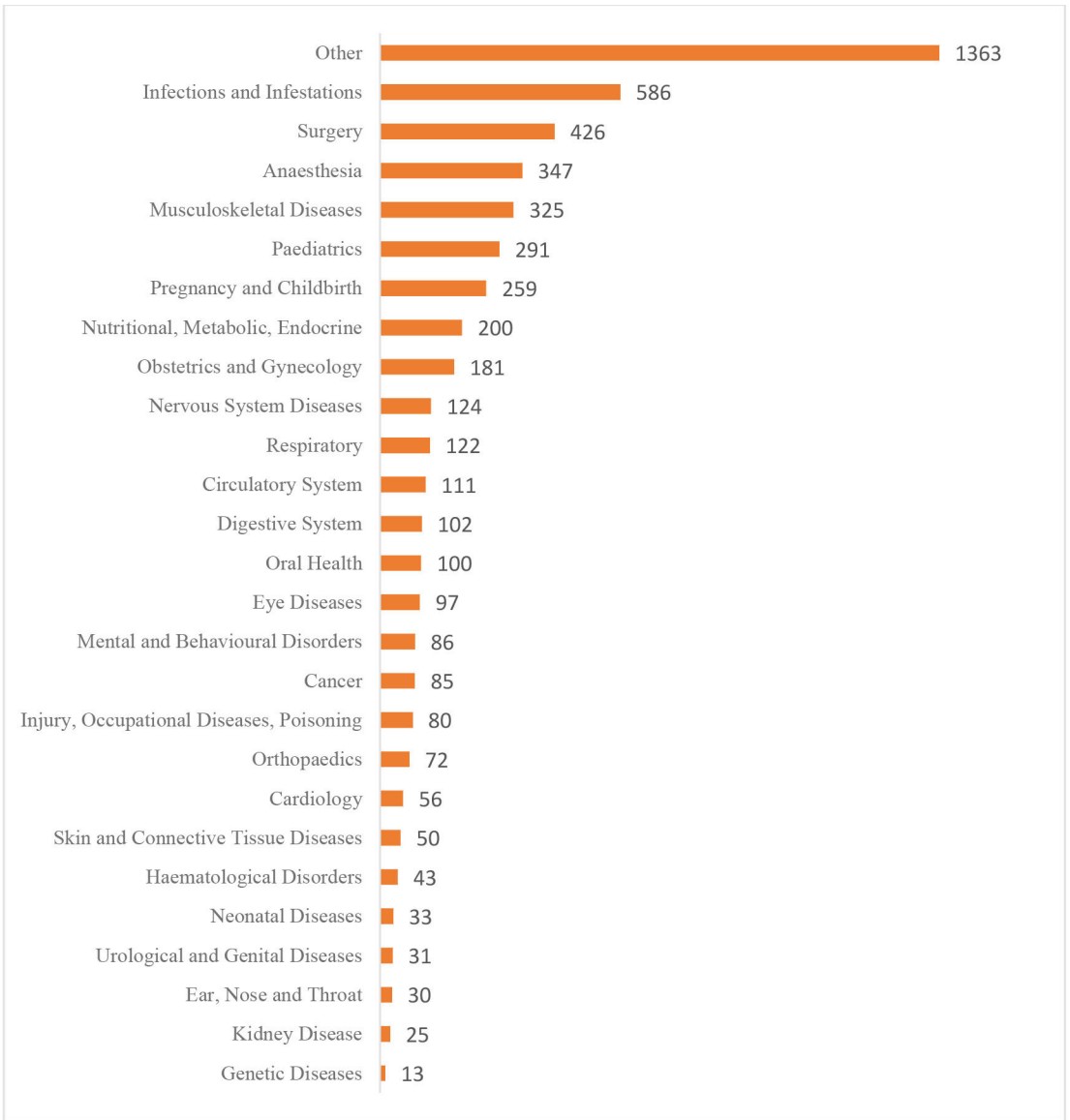

**Figure 4** Disease conditions investigated in PACTR-registered trials. An investigation of the disease categories is represented as bar graphs on the y axis. The number of trials registered to investigate a specific disease condition is presented on the bar. PACTR, Pan African Clinical Trials Registry.

funding agencies to ensure that results are in a public domain within a specified period[24] and that clinical trial reporting is not subjected to selective reporting and publication bias.[9 25 26]

The most common intervention in the trials conducted in Africa is treatment with drugs (21%), in which the trials registered in PACTR seek to find treatment options for infectious diseases (20%). This explains that in the most common diseases researched in Africa, there is a need for new drugs to curb the pandemics of these diseases.

The reporting section, item 24, suggests a trend towards being completed to conform to WHO-ICTRP requirements. Results reporting became mandatory in January 2019. Our analysis indicated an improvement in the reporting section completed. This trend continues with more trials in 2020 and 2021, opting for completing the reporting section. As part of our ongoing analysis,

this analysis shows that more needs to be done to build capacity on clinical trials through partnering with regulatory bodies, sponsors and researchers to ensure that clinical research conducted in Africa meets the global standards.

**Limitations**

We conducted a descriptive analysis of the data to analyse what is happening in the registry. We used the ICTRP platform to search for PACTR-registered studies. This approach may have excluded trials currently being processed at the time of downloading this data file which may be prior to sending monthly data file to ICTRP, thus under-representing the total number of registered trials. The disease category was too vast to understand the specific disease conditions investigated when analysing the data. In the future, we will focus on

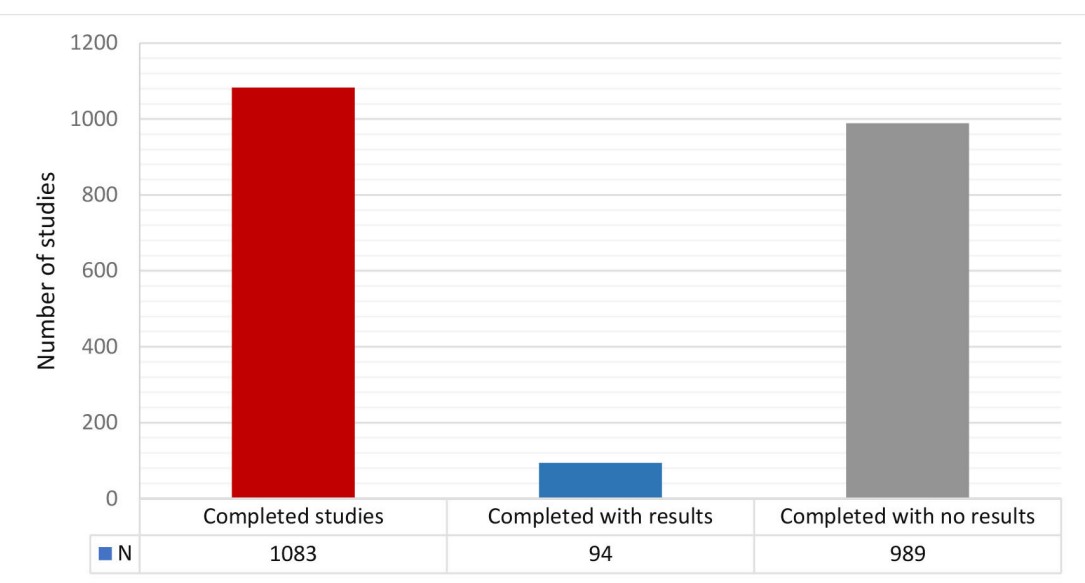

**Figure 5** Completed trials with results in PACTR. We describe the number of registered trials with a 'complete' status represented with the red bar and the actual number of the trials presented as N at the bottom of the bar. The blue bar represents the number of trials with available results, and the grey bar represents the completed trials without results. PACTR, Pan African Clinical Trials Registry.

unpacking the disease categories and understanding the trends in the diseases being evaluated. Also, there are data elements in which a researcher would indicate 'other', resulting in many trials with 'other' as a disease condition. Assessment of the data allows the PACTR review team to reinforce correct data entry when conducting reviews of the submitted trials and include

all mandatory data fields required by the WHO. Our analysis did not have the free-text data captured in the registry, which will need further unpacking to understand the trends of trial registration. The description of a sponsor into categories may limit what the researcher identifies as a sponsor to how we classified the sponsors leading to some variation.

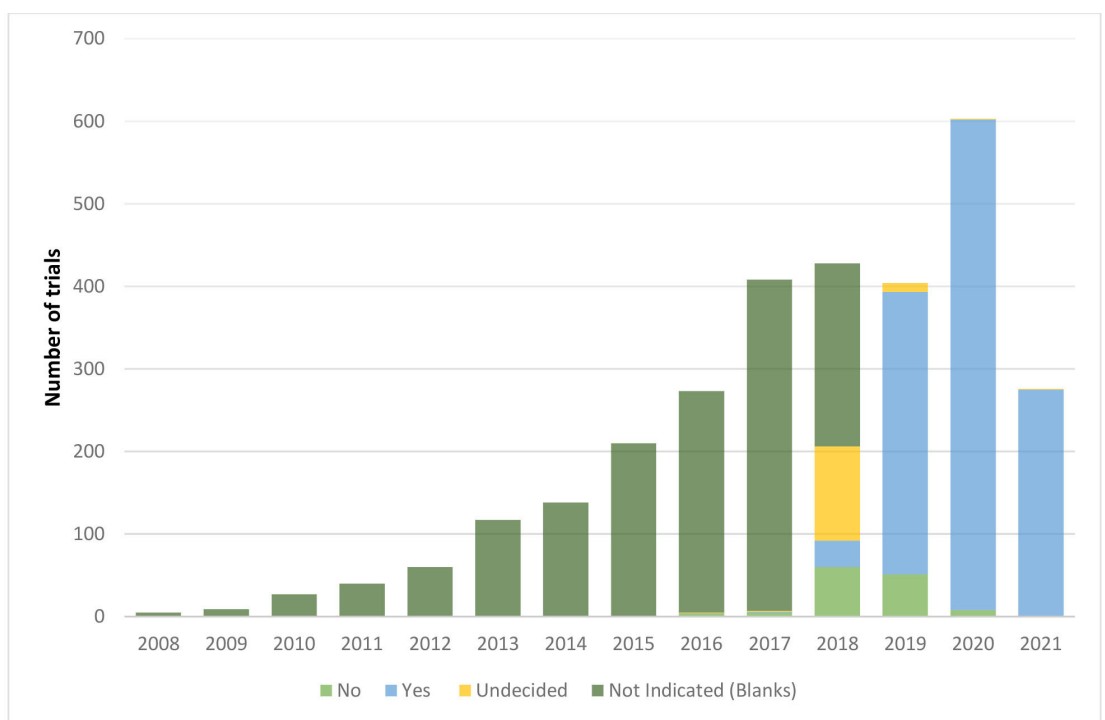

**Figure 6** Trends in results reporting over the years. We describe the reporting section of the registered trials from 2008 to 2021 presented in bars. The dark green bar indicates trial records in which the reporting section was not completed. The lighter green represents the trials that indicated 'no' to reporting results. Light blue indicates registered trials with results reported. The yellow bar represents the records of undecided results to report the results. The y axis is the total number of trials.

## Conclusion

Registration on PACTR has continued to grow since 2008. PACTR provides valuable data to map clinical trial conduct on the African continent. More work needs to be done to ensure that, as the registry team, we guarantee capacity building in collaboration with the ethics committee, funders and sponsors to provide that PACTR ensures that clinical research conducted in Africa meets international standards. There is an urgent need to continue to raise awareness for prospective trial registration and reporting of summary results. This will allow researchers to understand the importance of data sharing to contribute to research gaps to find solutions for Africa.

## Further considerations

PACTR should expand its efforts to build capacity in the African continent, explicitly creating links with ethics committees to evaluate the underlying quality of the scientific data included in these trials. We noted several instances where ethics documents submitted by the researcher needed to be verified to confirm that the ethics approval received is legit. Furthermore, research is required to understand the reasons for enforcing trial registration requirements by editors, funders and regulatory bodies across the continent.

**Acknowledgements** The authors acknowledge the South African Medical Research Council for funding support.

**Contributors** DEN wrote the first draft, coordinated and integrated comments from coauthors, approved the final version for publication and is the guarantor of the manuscript. SR and LM conducted the analysis of the data. SR, LM, EP, AH and CSW critically revised successive drafts of the manuscript, provided important intellectual input and approved the final version of the manuscript. The authors have read and approved the article's final version for submission.

**Funding** The publication cost for this article is supported with funds from the South African Medical Research Council (project code 43500).

**Disclaimer** The views expressed in this article are those of the authors. They do not necessarily reflect the opinions or policies of the South African Medical Research Council, Cochrane or other institutions that the authors are affiliated with.

**Competing interests** None declared.

**Patient consent for publication** Not required.

**Ethics approval** This study does not involve human participants.

**Provenance and peer review** Not commissioned; externally peer reviewed.

**Data availability statement** Data are available upon reasonable request.

**ORCID iDs**
Duduzile Edith Ndwandwe http://orcid.org/0000-0001-7129-3865
Lindi Mathebula http://orcid.org/0000-0002-4213-7318
Charles Shey Wiysonge http://orcid.org/0000-0002-1273-4779

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
