## [Reviewer comments · BMJ Open]

ARTICLE DETAILS

TITLE (PROVISIONAL)	Practices and trends in clinical trial registration in the Pan African Clinical Trials Registry (PACTR): a descriptive analysis of registration data
AUTHORS	Ndwandwe, Duduzile; Runeyi, Sinazo; Pienaar, Elizabeth; Mathebula, Lindi; Hohlfeld, Ameer; Wiysonge, Charles

VERSION 1 – REVIEW

REVIEWER	Ghassan, Karam Organisation mondiale de la Sante
REVIEW RETURNED	04-Oct-2021

GENERAL COMMENTS	Page 3 line 11: International Clinical Trials registry Platform (ICTRP) Page 7 line 24: the Trials in ICTRP include only registered trials. The last 2 sentences need to be reviewed, or clarified. It is not clear what the author mean by saying “which primary registered send data file...” Page 7 line 34: this paragraph is missing numbers that should describe each data set downloaded from both PACTR and ICTRP. Also, the sponsor field should specify if primary or secondary sponsor. The age range should be specified as age minimum and age maximum to be compatible with the description used by ICTRP. Page 8 line 14: “ICTRP is a one-stop portal...” Page 8 line 7: this first paragraph is confusing, it needs to be rewritten in order to clarify how many records are send to ICTRP and how many are not sent to ICTRP, both numbers constitute the total records of PACTR that are in the PACTR database. Page 8 line 24: this paragraph doesn’t mention anything related to COVID, it should say how many of the records registered in 2020 are COVID related and analyze those findings. It will be an added value to the paper.
--

REVIEWER	Giannuzzi, Viviana Fondazione per la Ricerca Farmacologica Gianni Benzi ONLUS
REVIEW RETURNED	18-Oct-2021

GENERAL COMMENTS	Dear authors, Abstract: - a few more details on the database should be provided, e.g. that the registration is not compulsory, it is linked to the WHO one, and if it covers all African countries. what ethics does it mean (ethics approval received?) unclear: 3000 marks (n=2998).
---

	Introduction:  - not clear if the database is freely available. If yes, the relevant links of the database should be provided - given that an important result is the obtainment of the ethical approval, in the introduction it should be specified if the ethics approval is mandatory for interventional trials and where Material and methods:  - the relevant link to the public databases should be provided. Results: there are several missing explanations, maybe to be provided in the methods section  - it is not clear what 'not registered' means in the sentence "Of these records, we excluded 1964 that were not registered." - the sentence "The data sent to WHO ICTRP includes both registered trials and those pending, which can be linked directly to the registry that sent the data" is not clear, as well. - Figure 1: it is not clear why data have been stratified in 'retrospective' and 'prospective'. In addition, these two adjectives could be misleading because clinical studies can be retrospective and prospective in the collection of samples data and/or samples (other than interventional clinical trials that are always prospective). It is neither explained why this classification. Given that this data is provided in the Table 1, I would suggest to avoid in this figure such a stratification. - it is explained what "factorial assignment" means - Table 1 reports that there are no "Ongoing/Active" trials. This findings is not discussed/explained. - Figure 3: it is strange to find 'surgery' as a 'condition': please explain/amend - Figure 4: the first column is maybe useless - the sentence "this field was not mandatory and had options "yes" "no" and "undecided." could be fully explained as well as the following text commenting figure 5. It is also unclear the intent of providing data from figure 5. Discussion in general, each topic should be introduced and explained. for example, the sentence "Trials are allowed to be registered retrospectively; however, prospective registration of trials is promoted" should be further discussed- the "intervention model" should be explained. what does it refer to? Finally, more considerations from authors should be added. Limitations: are too vague and miss some important information.
--	---

VERSION 1 – AUTHOR RESPONSE

Reviewer: 1

Mr. Karam Ghassan, Organisation mondiale de la Sante

Comments to the Author:

Page 3 line 11: International Clinical Trials registry Platform (ICTRP)

Response: Thank you for the comment. We have checked and verified that the wording is correct on page 3; line 25.

Page 7, line 24: the Trials in ICTRP include only registered trials. The last 2 sentences need to be reviewed or clarified. It is not clear what the author mean by saying “which primary registered send data file...”

Response: We have taken the opportunity to go back and verify the dataset used for the analysis, thus rectifying the confusing statement, thank you. Please refer to pages 9-10

Page 7 line 34: this paragraph is missing numbers that should describe each data set downloaded from both PACTR and ICTRP. Also, the sponsor field should specify if primary or secondary sponsor. The age range should be specified as age minimum and age maximum to be compatible with the description used by ICTRP.

Response: We only used the dataset from ICTRP and only reported on the primary sponsor as listed. We have amended table 1 and also updated the description of the dataset as well as the data fields, which we analysed on pages 9-10

Page 8 line 14: “ICTRP is a one-stop portal...”

Response: Thank you, we have corrected the statement accordingly on page 9; line 154

Page 8 line 7: this first paragraph is confusing, it needs to be rewritten in order to clarify how many records are send to ICTRP and how many are not sent to ICTRP, both numbers constitute the total records of PACTR that are in the PACTR database.

Response: We have taken the liberty of revisiting the data set and clearly specifying the database used for the analysis as described in the Methods section on pages 9-10

Page 8 line 24: this paragraph doesn't mention anything related to COVID, it should say how many of the records registered in 2020 are COVID related and analyze those findings. It will be an added value to the paper.

Response: Thank you, we have added a few details of the COVID-19 trials for 2020 on pages 11-12 and figure 2. We have a separate manuscript which is focusing on COVID-19 trials.

Reviewer: 2

Dr. Viviana Giannuzzi, Fondazione per la Ricerca Farmacologica Gianni Benzi ONLUS

Comments to the Author:

Dear authors,

Abstract:

- a few more details on the database should be provided, e.g. that the registration is not compulsory, it is linked to the WHO one, and if it covers all African countries.

what ethics does it mean (ethics approval received?)

unclear: 3000 marks (n=2998).

Response: Thank you for the comment, we have added the details accordingly on pages 3-4. The PACTR registry is a WHO primary register with a remit for Africa. Ethics approval is an ethical requirement for any clinical trial involving human participants. We have analysed the trial records in the registries to understand if whether ethics approval has been received for the records in PACTR. Additionally, we have revised the unclear statements

Introduction:

- not clear if the database is freely available. If yes, the relevant links of the database should be provided

Response: Thank you for the suggestion, we have provided the relevant links in the background

- given that an important result is the obtainment of the ethical approval, in the introduction, it should be specified if the ethics approval is mandatory for interventional trials and where
Response: Thank you. The comment has been addressed in the background section on page 6-8, , line 111-112.

Material and methods:

- the relevant link to the public databases should be provided.

Response: Thank you, we have included the links to the respective databases on page 9

Results

there are several missing explanations, maybe to be provided in the methods section

- it is not clear what 'not registered' means in the sentence "Of these records, we excluded 1964 that were not registered."

- the sentence "The data sent to WHO

ICTRP includes both registered trials and those pending, which can be linked directly to the registry that sent the data" is not clear, as well.

Response: We have taken note of this comment and have corrected it accordingly on page 9; lines 152-177-

- Figure 1: it is not clear why data have been stratified in 'retrospective' and 'prospective'. In addition, these two adjectives could be misleading because clinical studies can be retrospective and prospective in the collection of samples data and/or samples (other than interventional clinical trials that are always prospective). It is neither explained why this classification. Given that this data is provided in the Table 1, I would suggest to avoid in this figure such a stratification.

Response: Thank you for this comment. To clarify, we have the stratification based on how clinical trials are flagged in the registry, and figure 1 was to highlight the trends in the retrospective vs prospective trial registration. Clinical trial registers advocates for prospective trial registration, and as such, we wanted to show that the trend in the trials registered in PACTR is shifting towards prospective trial registration of trials.

- it is explained what "factorial assignment" means

Response: The explanation of factorial assignment had been added on page 11; line 237-238. Thank you for the comment.

- Table 1 reports that there are no "Ongoing/Active" trials. This findings is not discussed/explained.

Response: We have corrected the wording in table 1 to reflect the explanation given on page 13; line 220

- Figure 3: it is strange to find 'surgery' as a 'condition': please explain/amend

Response: Thank you for the comment. We have provided an explanation on page 16; lines 259-262

- Figure 4: the first column is maybe useless

Response: Thank you for the comment, ,however, see the importance of indicating the trials that have been completed to make a clear case that among the completed studies, a small proportion has reported their results in the registry.

- the sentence "this field was not mandatory and had options "yes" "no" and "undecided." could be fully explained as well as the following text commenting figure 5. It is also unclear the intent of providing data from figure 5.

Response: Thank you for the comment. The addition of the data fields has led to more trials opting to share their summary results. The explanation is added on page 18; lines 277-278

Discussion

in general, each topic should be introduced and explained.

for example, the sentence "Trials are

allowed to be registered retrospectively; however, prospective registration of trials is promoted" should be further discussed- the "intervention model" should be explained. what does it refer to?

Response: Thank you. We have provided the explanations and wish to express that clinical trial registrations in general endeavor to promote prospective trial registration in the registry. We have provided further clarification on this topic on page 7; lines 99-114. Also, the intervention model has been defined on page 13; line 223

Finally, more considerations from authors should be added.

Response: Thank you. We have taken note of the comment and wish to point out that considerations by the authors are expressed in the discussion and conclusion section on pages 20-23

Limitations: are too vague and miss some important information.

Response: We thank you for the comment and have improved the limitation section on page 22; lines 256 - 372

VERSION 2 – REVIEW

REVIEWER	Ghassan, Karam Organisation mondiale de la Sante
REVIEW RETURNED	05-Dec-2021

GENERAL COMMENTS	line 132: missing "by" line 142: what was downloaded from the ICTR? not clear, maybe the word data or records is missing there. in that same paragraph, the use of the present tense in the beginning and then past tense is confusing and done without any transition. line 215: "PACTR saw in 2020 had registered 606 trials..." is not clear.
--

REVIEWER	Giannuzzi, Viviana Fondazione per la Ricerca Farmacologica Gianni Benzi ONLUS
REVIEW RETURNED	06-Dec-2021

GENERAL COMMENTS	The abstract still misses information on the compulsory (or not) nature of the register In the introduction, please clearly state in the text that the database is freely available. if the authors would like to keep figure 1 on the basis of the clarifications given, this should be specified in the text. as said, these two adjectives could be misleading because clinical studies can be retrospective and prospective in the collection of samples data and/or samples (other than interventional clinical trials that are always prospective). It is neither explained why this classification.
--

	further considerations in the discussion have not been provided
--	---

VERSION 2 – AUTHOR RESPONSE

Reviewer: 1

Mr. Karam Ghassan, Organisation mondiale de la Sante

Comments to the Author:

*line 132: missing "by"

Response: Thank you for noting our omission, we have corrected the wording to include "by" in line 132 of the track change version

*line 142: what was downloaded from the ICTR? not clear, maybe the word data or records is missing there. in that same paragraph, the use of the present tense in the beginning and then past tense is confusing and done without any transition.

Response: Thank you for the comment, we have amended appropriately on line 132 of the track change version

*line 215: "PACTR saw in 2020 had registered 606 trials..." is not clear.

Response: Thank you for the comment. We have rectified the statement to make it clear on line 215-216 of the track change version

Reviewer: 2

Dr. Viviana Giannuzzi, Fondazione per la Ricerca Farmacologica Gianni Benzi ONLUS

Comments to the Author:

*The abstract still misses information on the compulsory (or not) nature of the register

Response: We wish to point out that clinical trial registration is recommended to conform to the International Committee of Medical Journal Editors (ICMJE) for prospective trial registration (<http://www.icmje.org/recommendations/browse/publishing-and-editorial-issues/clinical-trial-registration.html>). In some ways, a researcher would need to register in order to have their trials results published because other journals would want the manuscript to indicate the registry number.

*In the introduction, please clearly state in the text that the database is freely available.

Response: Thank you for the comment, we have added the statement to indicate that it is freely available on line 74 of the track change version

*If the authors would like to keep figure 1 on the basis of the clarifications given, this should be specified in the text. As said, these two adjectives could be misleading because clinical studies can be retrospective and prospective in the collection of samples data and/or samples (other than interventional clinical trials that are always prospective). It is neither explained why this classification.

Response: Thank you for the comment. We wish to mention that the registration process of a trial includes the status “retrospective” or “prospective” registration which would indicate to the journal whether the trial was registered prospectively or retrospectively. This status refers to the registration and not collection of data. To ensure transparency, it is recommended that clinical trials register their trials prospectively.

*Further considerations in the discussion have not been provided

Response: Thank you for the comment. We have added the further considerations section after the discussion section on page 16, line 290 -296 of the track change version

VERSION 3 – REVIEW

REVIEWER	Giannuzzi, Viviana Fondazione per la Ricerca Farmacologica Gianni Benzi ONLUS
REVIEW RETURNED	05-Jan-2022
GENERAL COMMENTS	I would suggest to indicate the non-compulsory nature of the register in the abstract.